The use of synthetic agonists of quorum sensing N- acyl homoserine lactone pathway improves the bioleaching ability in Acidithiobacillus and Pseudomonas bacteria

Caicedo Juan Carlos jua.caicedo@mail.udes.edu.co 1
Villamizar Sonia 2
Orlandoni Giampaolo 1
1 Science Faculty, Universidad de Santander , Bucaramanga , Santander , Colombia
2 School of Agricultural and Veterinarian Sciences, Universidade Estadual Paulista , Jaboticabal , São Paulo , Brazil
Gillespie Joseph
Electronic publication date: 2022 Aug 9
Publication date: 2022
Volume: 10
Electronic Location ID: e13801
Received 2022 Jan 24; Accepted 2022 Jul 6
Copyright: ©2022 Caicedo et al.
Copyright year: 2022
Copyright holder: Caicedo et al.
License: This is an open access article distributed under the terms of the Creative Commons Attribution License, which permits unrestricted use, distribution, reproduction and adaptation in any medium and for any purpose provided that it is properly attributed. For attribution, the original author(s), title, publication source (PeerJ) and either DOI or URL of the article must be cited.
License URL: https://creativecommons.org/licenses/by/4.0/

Keywords: Biofilm, Auto-inducer, Hydrogen cyanide, Bioleaching, Cell attachment

Funding: Administrative Department of Science and Technology MINCIENCIAS Energy Mining Planning Unit UPME 80740-008-2020 Universidad de Santander This research was funded by the Administrative Department of Science and Technology MINCIENCIAS and Energy Mining Planning Unit UPME, Grant No 80740-008-2020 and the APC was funded by Universidad de Santander. The funders had no role in study design, data collection and analysis, decision to publish, or preparation of the manuscript.

==============================
Metal solubilization from discarded electrical material and electronic devices (e-waste) using the bioleaching capabilities of bacterial cells is highly effective. However, gaps in understanding about the microbiological processes involved in the bioleaching reaction leads to less efficient metal solubilization in large-scale e-waste processing. In this study, bacterial species belonging to the genera Acidithiobacillus and Pseudomonas were used to leach copper and gold from discarded printed circuit boards (PCB). Through modulation of the cell-to-cell communication system in these bacteria, phenotypic traits directly involved in the bioleaching reaction were regulated in order to improve the metal solubilization. Addition of the long chain synthetic autoinducer molecule N-acyl homoserine lactone (AHL) of the quorum sensing pathway to the bioleaching reaction resulted in a significant enhancement of metal extraction from PCB. Factors such as: cell attachment to PCB, biofilm formation and hydrogen cyanide (HCN) production were regulated by the quorum sensing system and could be directly related to the improvement of metal bioleaching. Bioleaching reactions using bacterial quorum sensing modulation could represent a valuable tool in overcoming limitations at the industrial level imposed by microbiological traits that lead to inefficient metal bioleaching from e-waste.

Introduction

Due to the rapid rise in the world population, the demand on natural resources is increasing. One major concern is the rapidly declining metal ore deposits. Currently, electronics and electrical devices (EED) are the products with the highest demand worldwide. The progressive reduction in production costs and their relatively short lifespan stimulates frequent purchasing of brand-new EED. This activity contributes to the depletion of several metals such as: copper, platinum, silver, palladium and gold, which are essential in EED manufacturing (Mihai et al., 2019), in addition to causing a 5-fold upsurge of electric and electronic waste generation (e-waste) (Robinson, 2009). Developed countries have developed recycling policies; however, a substantial amount of e-waste is transported legally or clandestinely to developing countries where this e-waste is dumped in landfills or incinerated (Greenpeace, 2008). The e-waste has been accumulating for years; however, only in the last few years has the enormous risk to the environment and human health become apparent. The core piece of EED is the printed circuit board (PCB). PCB contains several organic and inorganic pollutants (i.e., metals such as mercury, cadmium, chromium, copper, and persistent organic pollutants such as dibenzofurans, polychlorinated biphenyls, and polycyclic aromatic hydrocarbons) (Xinhui et al., 2010). On the other hand, the content of precious and semi-precious metals in PCB are hundreds fold more than in a natural ore. Consequently, the PCBs are considered a source of metals, for that reason chemical (hydrometallurgy) and physical (pyrometallurgy) approaches have been implemented in order to recover these metals. The main disadvantage of these approaches are: the hydrometallurgy recovery method uses large amounts of acidic or caustic solutions in order to dissolve the material, which leading to corrosive and/or toxic wastewater accumulation in the environment (Hoffmann, 1992). The pyrometallurgy approach consumes large amounts of energy in order to smelt the material and produces several gas pollutants such as dioxins and furans (Sum, 1991).

Biohydrometallurgy is a low investment and environmentally friendly approach that has been implemented in the last decade. This method uses bacteria and fungus as bioleaching agents in order to solubilize metals such as copper, gold and zinc (Brandl, Bosshard & Wegmann, 2001). The bacterial genus Acidithiobacillus is usually recognized for its bioleaching potential. Bacterial species belonging to this genus are frequent inhabitants of sulfur containing acidic environments. Such as acid mine drainage, iron-sulfur mineral mines and hot springs (Chen, Lin & Lin, 2021). The species from Acidithiobacillus genus have a circular genome with 3 MB in size and total of 3,217 protein coding genes, as well characteristic genetic determinants with potential pathogenic effect on humans have not been identified in this bacterial genus. (Valdés et al., 2008) A. ferrooxidans and A. ferrivorans are studied for their ability to bioleach copper. These bacteria are acidophiles and chemoautotrophs, their main energy source comes from the oxidation of iron Fe2+ and sulfur complexes (Hedrich & Johnson, 2013). The mechanisms of Cu leaching by Acidiothibacillus sp. involve indirect leaching by biogenic sulfuric acid oxidizing elemental S0 to H2SO4. Fe2+ acts as an electron donor and is oxidized to Fe3+, which then acts as the oxidizing agent, catalyzing the leaching reaction (7). Bioleaching reactions depend on the initial pH, Fe2+ concentration and oxidation rate. The main bottleneck of biohydrometallurgy is the restriction for large scale implementation, which could be a consequence of the incomplete understanding of the main microbiological features involved in the bioleaching process.

Quorum sensing (QS) is a cell to cell communication system widely distributed in bacteria. The QS system is based on production, secretion and perception of signaling molecules termed autoinducers. Since QS controls the expression of nearly 25% of non-essential genes, a deeper understanding of this system and subsequent modulation in bacteria involved in biohydrometallurgy could enhance bioleaching reactions in e-waste at the industrial level (Caicedo & Villamizar, 2021). Cell attachment and subsequent biofilm formation are critical prerequisites for metal solubilization by Acidiothiobacillus sp, these phenotypic traits are regulated by a cluster of genes involved in quorum sensing system. Bacterial species belonging to the Acidithiobacillus genus possess two quorum sensing systems mediated by the autoinducer molecule AHL: the AfeI/R and ACT (acyl transfer function) quorum sensing systems (Schaefer et al., 2008; Rivas et al., 2007). Other bacteria with well-known gold bioleaching potential belong to Pseudomonas genus (Shin et al., 2013). Pseudomonas genus is characterized by a great environmental ubiquity not only due to the extraordinary ability of members to produce a wide diversity of secondary metabolites as well its great genome plasticity (Villamizar et al., 2020). P. putida is a frequent rhizosphere and freshwater inhabitant, infections caused by these bacteria are unusual, however its isolation has been reported from patients with some degree of immunosuppression (Fernandez et al., 2015). P. fluorescens is bacterium commonly isolated from rizospheres and is widely recognized but its ability as plant growth promoting rhizobacteria activity (Villamizar, 2021). These bacterial species produce cyanide as a secondary metabolite in the process of glycine oxidative decarboxylation, this biosynthetic cyanide has a pivotal role in gold bioleaching (Işıldar et al., 2016). In the Pseudomonas genus, the transcriptional control of genes encoding the enzyme HCN synthase, which catalyzes the production of HCN, is modulated by two quorum sensing circuits: LasI/LasR and Rh1I/Rh1R, which both use AHL as an auto-inducer molecule (Pessi & Haas 2000). In this study, the effect of the quorum sensing system on the bioleaching reaction was tested in vitro, in a two-step bioleaching approach, using synthetic agonists of the auto-inducer molecule AHL family in bioleaching reactions of Cu and Au from PCB, the bioleaching agents used were: A. ferrooxidans and A. ferrivorans for copper, and P. putida and P. fluorescens for gold.

Materials & Methods

E-waste source and crushing conditions

Discarded electronic devices, namely desktop computers, laptops, mobile phones, printers and monitors, were collected by UDES Verde (recycling program of the University of Santander). The cast-off devices were manually disassembled and the PCBs were removed. The PCBs were grouped into the following categories: desktop computer boards, desktop computer boards without components, laptop boards, mobile phone boards, printer boards and keyboard boards. Batteries, heat sinks, and PCB microprocessors were removed from desktop computers. Board components, such as connectors, capacitors, and onboard chips, were removed using specialized hardware equipment. The PCBs were then washed with distilled water, manually cut into ∼3 cm × 3 cm pieces (Fig. 1A), and then ground on a cutting mill (Fritsch Pulverissette 19), using a 2.5 mm trapezoidal sieve. The grinding conditions were: (i) Pre-crushed stage: the 3 × 3 cm size pieces were crushed at 500 RPM for 15 min to obtain a final particle size of one mm, (ii) crushed stage: afterwards, the pre-crushed material was ground at 3,000 RPM, for 5 min in order to obtain final particle sizes of 1,000, 500 and 250 µm (Fig. 1B). The ground PCB was washed with distilled water and dried overnight prior to determination of the initial concentration of metals (Cu and Au). The USEPA 3052 method (microwave assisted acid digestion with HCl and HNO33:1 v/v) was used to digest the siliceous and organic components from PCB. Initial metal content determination assays were performed by inductively coupled plasma mass spectrometry (SQ-ICP-MS; Thermo Scientific, Waltham, MA, USA). For the subsequent bioleaching tests, the ground PCB material was sterilized by autoclaving for 15 min at 121 °C, and 15 psi.

Figure 1 PCB disassembled and crushed.

(A) PCBs disassembled, 3 × 3 cm manually cut pieces. (B) PCB crushed for bioleaching assays final size 250 µm.

Bacterial strains and culture media

The bacteria used in this study were commercially acquired from German Collection of Microorganisms and Cell Cultures GmbH (DSMZ), Germany, including the acidophile strains; Acidithiobacillus ferrivorans (DSM 17398) and Acidithiobacillus ferrooxidans (DSM 9464), as well as cyanide-producing strains; Pseudomonas putida (DSM 10234) and Pseudomonas fluorescens (DSM 106119). A. ferrooxidans and A. ferrivorans were recovered in DSM 882 mineral medium containing: (NH4)2SO4 (2.0 g/L−1), MgSO4 ⋅ 7H2O (0.25 g/L−1), KH2PO4 (0.1 g/L−1), KCl (0.1 g/L−1), FeSO4 ⋅ 7H2O (8.0 g/L−1) and S0 (10.0 g/L−1), the pH was adjusted to 2.5 with sulfuric acid. Subsequently, the mineral medium containing the acidophilic bacteria was incubated at 30 °C at aerobic conditions until an apparent growth was obtained. In order to recover the Pseudomonas strains, the bacteria were grown in selective and differential microbiological medium King’s B Agar containing: 20 g/L−1 peptone, 1.5 g/L−1 dipotassium hydrogen phosphate, 1.5 g/L−1 magnesium sulfate, 15 g/L−1 agar and 5- chloro-2 - (2,4-dichlorophenoxy) phenol 25 mg/L−1 and King’s A modified agar containing: peptone 20 g/L−1, dipotassium hydrogen phosphate 1.5 g/L−1, magnesium chloride 1.5 g/L−1, agar 15 g/L−1 and 5- chloro-2- (2,4-dichlorophenoxy) phenol 25 mg/L−1. Plates were incubated at 28 °C.

Copper bioleaching assays

A two-stage bacterial growth approach was used in the Cu bioleaching assays, to allow the establishment and adaptation of Acidithiobacillus cells, as well as to maintain optimal bioleaching conditions. In the first stage (pre-leaching), five mL of pure and mixed cell suspensions of A. ferrooxidans and A. ferrivorans at a final concentration of 106 CUF/mL−1 were added to 95 mL of 882 DSM media culture containing FeSO4 at 20 and 10%. Subsequently, they were incubated for 7 days at 30 °C with 180 RPM shaking. No e-waste was added at this first stage. In the second step (leaching): 0.5, 1, 2.5 and 5 g of sterile ground PCB were added to 100 mL of 882 DSM media containing active bacteria at a concentration of 1.5 × 108 CFU/mL−1, measured at OD600. Culture conditions were: 28 °C, 180 RPM and 480 h. Control assays were performed with the same culture conditions without a bacterial inoculum. The pH and ORP (oxidation reduction potential) were measured every 12-hour using the portable HI98191 pH/ORP/ISE meter (Hanna Instruments, Woonsocket, RI, USA). Cu concentration in the leachate solution was measured every 24 h for 28 days by AAS (atomic absorption spectroscopy) at 324.7 nm using the ICE™ 3500 (Thermo Scientific). All assays were performed in triplicate, and the statistical analysis of data was performed using one-way ANOVA with post hoc testing (Bonferroni) using SPSS STATISTICS DESKTOP, v. 22.0 software (IBM).

Gold bioleaching assays

The bioleaching agents used in these assays were the cyanogenic strains P. putida and P. fluorescens. Semiquantitative and quantitative assays based on HCN synthesis by the oxidative decarboxylation of glycine, were performed in order to determine the point of highest HCN production by the strains (Fernandez et al., 2015). The picrate method was performed to measure the HCN production. This method is based on the formation of a red compound namely “isoporpuric acid”, which is produced by the reaction of HCN and alkaline picrate (Bakker & Schipper, 1987). In brief, the bacteria were plated in King’s B Medium supplemented with different concentrations of glycine (4, 8 and 12 g/L−1). Sterile filter paper soaked in a saturated alkaline picrate solution was placed in the upper lid of the petri dishes. Then, the plates were sealed with parafilm and incubated at 30 °C for 48 h. The change of color on the filter paper denoted the HCN synthesis. The scale of results based on the color changes was: yellow to light brown signified weak HCN synthesis, brown signified moderate HCN synthesis and brick color was registered as strong HCN synthesis (Willians & Edwards, 1980). For quantitative assays, the bacteria were grown in King’s B broth and sterile paper strip soaked in alkaline picrate solution was inserted into the broth, after 48 h of incubation at 30 °C, the color in the paper was eluted with water and its absorbance was read at 625 nm. All assays were performed in triplicate.

In our two-step bioleaching approach, the remaining PCBs were collected from the Cu bioleaching reaction, allowed to dry overnight, and subsequently sterilized with two rounds of autoclaving for 15 min each at 121 °C, and 15 psi. 1g of this material was then added to 100 mL of NB medium containing the actively growing cells of P. putida and P. fluorescens at the point of highest cyanide production; the medium was also supplemented with 4, 8 and 12 g/L−1 of glycine. The bacteria were then cultivated at 28 °C with 160 RPM orbital shaking for 120 h. Control assays were performed with same conditions without a bacterial inoculum. Au concentration in the leachate solution was measured every 12 h by atomic absorption spectroscopy at 242.8 nm using the ICETM 3500 (Thermo Scientific). All assays were performed in triplicate, and the statistical analysis of data was performed using one-way ANOVA with post hoc testing (Bonferroni) using SPSS STATISTICS DESKTOP, v. 22.0 software (IBM).

Synthetic agonist of quorum sensing implementation

As mention above, the bioleaching agents used in this study, the bacteria A. ferrivorans, A. ferrooxidans, P. putida and P. fluorescens share a quorum sensing system mediated by the auto-inducer molecule family AHL. In order to determine the effect of synthetic AHL analogues on the phenotypic traits, it which favor the bacterial leaching reactions in all strains used in this study. Diverse synthetic analogues listed in Table 1 were added at a final concentration of 5 µM in DMSO to the bacterial cultures in the bioleaching shake flask. The culture conditions were the same to for both Cu and Au bioleaching assays, which are described above. Periodic measurements of Cu and Au concentration in the leachate were performed by AAS. Bioleaching reactions with the same incubation conditions without AHLs analogues were used as control test. All assays were performed in triplicate.

Table 1 Synthetic analogues of AHL autoinducer molecule used in this study.

All analogues were commercially obtained from sigma Aldrich ≥ 97.0 HPLC.

Synthetic analogue	Empirical formula	Molecular weight	Structure	QS bacterial strain	
N-(3-oxododecanoyl)-L-homoserine lactone	C16H27NO4*	297.39		P. putida and P. fluorescens	
N-Decanoyl-L-homoserine lactone	C14 H25 NO3*	255.35		P. putida and P. fluorescens	
N-(3-Hydroxytetradecanoyl)-DL-homoserine lactone	C18 H33 NO4*	327.46		A. ferrooxidans and A. ferrivorans	
N-Dodecanoyl-DL-homoserine lactone	C16 H29NO3*	283.41		A. ferrooxidans and A. ferrivorans,	
Notes.

* The final concentration for assays was 5 µM.

Biofilm formation assessment by crystal violet retention assay

This assay was performed in order to determine whether the synthetic analogues of AHL have an effect on the bacterial adhesion to ground PCB and the subsequent biofilm development , as well if this attachment plays a role in Cu bioleaching. After measuring the Cu concentration in the leachate, the biofilm formation on ground PCB by A. ferrooxidans and A. ferrivorans was determined following the crystal violet retention by peptidoglycan layer method at 562 nm (Zhao, Chen & Nan, 2015) with modifications. Briefly, the tests were carried out in a 300 mL erlenmeyer flask containing 100 mL of 882 DSM with S0 (0.8% w/v) as an energy source, and the AHL synthetic analogues C18H33NO4 and C16H29NO3. at a final concentration of 5 µM. Then, actively growing strains of A. ferrooxidans and A. ferrivorans were added at a concentration of 1.5 ×108 CFU/mL−1. 1 g of ground PCB was then added, and three particle sizes: 1,000, 500 and 250 µm were assayed. The culture conditions were: 28 °C, 150 RPM and 72 h. Subsequently at 72 h post incubation, the ground PCBs were recovered from the bioleaching reaction flask by centrifugation. Afterward, 0.7 g of ground PCBs were deposited in 1.5 mL Eppendorf tubes, and stained with 1% crystal violet for 15 min at room temperature. To eliminate non-adherent cells and extra stain, it was washed 3×with tap water. To solubilize the crystal violet retained by the bacterial cells adhered to the pcb, which that make up the biofilm formed, 200 µL of 95% ethanol was added and its absorbance was measured at 562 nm. In order to determine if the bacterial attachment have effect over bioleaching reaction The Cu concentration in the leachate was measured at 24, and 72 h by AAS.

Bioleaching reactions with the same conditions without AHLs analogues were used as a control. All assays were performed in triplicate.

Data analysis

According to our previous study (Villamizar et al., 2020), all test were performed in triplicate and repetitions in independent trials as minimum three times were done. All data including in charts represent mean values. Bars ± represent the standard deviation of measurements. One-way ANOVA with post hoc test (Bonferroni) was performed by SPSS STATISTICS DESKTOP, v. 22.0 software (IBM). P < 0.05 was accepted for all analyses.

Results

Copper bioleaching by acidophile strains

The ability of A. ferrooxidans and A. ferrivorans to mobilize Cu from discarded PCB using a pre-leaching approach, as well as the effect of AHL quorum sensing system on their bioleaching capacity were analyzed. The initial Cu content in PCB sample measured by inductively coupled plasma mass spectrometry was 230 mg/g of crushed PCB. The bioleaching effectiveness of A. ferrivorans and A. ferrooxidans at 1% ground PCB pulp density were 95 and 91% respectively (Fig. 2). Additionally, 96% of the Cu was mobilized from the PCB by a mixed culture of the two acidophile strains. Higher loads of ground PCB substantially decreased the bioleaching ability of all the strains assayed (Fig. 3). The Cu mobilization efficiency of all acidophile strains is greatly reduced when the ground PCB added to the bioleaching reaction is higher than 1%. An insignificant Cu bioleaching effect was observed in a control (not inoculated) assay, which could be explained by the very low pH in the medium.

Figure 2 Cu Bioleaching patterns and pH using ground PCB at 1%.

Each time point represents the average of three replicate samples. Three replicates were used in three independent experiments displayed similar results.

Figure 3 Cu Bioleaching patterns using different pulp densities of ground PCB at 20 days of incubation.

Each bar represents the average of three replicate samples. Three replicated were assayed in three independent experiments.

Addition of AHL synthetic analogues improve the bioleaching efficiency at higher load of PCB

The effect of long chain AHL synthetic analogues on Cu bioleaching efficiency in acidophile bacteria was tested. We found substantial differences between bioleaching reactions treated with AHL synthetic analogues (C18 H33 NO4 and C16 H29NO3) and those not treated. This effect was much more apparent at a higher concentration of ground PCB (Fig. 4). The addition of C16H29NO3 to the bioleaching reaction by A. ferrivorans and A. ferrooxidans in either single culture or mixed culture enhanced the Cu mobilization from discarded PCB by 0.6 fold. In contrast, the bioleaching reaction supplemented with C18H33NO4 caused a slight reduction in Cu mobilization compared to the control assay with no AHL molecules. Although a small improvement was found in the Cu bioleaching ability of A. ferrivorans and A. ferrooxidans when using a PCB load of 5%, this difference was not significant when compared with the control (data not show).

Figure 4 Effect of AHL synthetic analogues over Cu bioleaching efficiency by A. ferrivorans and A. ferrooxidans.

(A) N-dodecanoyl-DL-homoserine lactone (C16H29NO3) was added at concentration of 5 µM. (B) N-(3-Hydroxytetradecanoyl)-DL-homoserine lactone (C18H33NO4) was added at concentration of 5 µM. Control assays with not AHL synthetic analogues molecules. Comparable patterns of Cu bioleaching were obtained in three independent experiments for every pulp density of PCB tested.

AHL synthetic analogues promote the biofilm formation leading to a higher bioleaching efficiency

We assessed the effect of the AHL synthetic analogues C16H29NO3 and C18H33NO4 on biofilm development on ground PCB. Bioleaching reactions treated with C16H29NO3 showed a significant biofilm formation by A. ferrivorans and A. ferrooxidans on ground PCB, predominantly when the particle size of ground PCB was below 500 µm. Additionally, samples with a high biofilm formation showed an improvement in Cu bioleaching activity by acidophile strains (Fig. 5). Treatment with C18H33NO4 did not cause significant differences in biofilm formation when compared with the control that did not contain AHL synthetic analogues (Data not shown)

Figure 5 Assays of biofilm formation on ground PCB and pH by A. ferrooxidans and A. ferrivorans treated with C16H29NO3 and its effect in Cu bioleaching ability.

Measurements made at 24 h post-treatment. (A) Ground PCB at 1,000 µm of particle size, (B) ground PCB at 500 µm of particle size and (C) ground PCB at 250 µm of particle size. Each bar and line represent the average of three replicate samples. Three replicates were used for every pulp density of PCB tested.

Gold bioleaching by Pseudomonas strains

The ability to produce biogenic cyanide was verified in the two Pseudomonas strains, using the picric acid method. Pseudomonas putida showed a slightly higher cyanide production compared to Pseudomonas fluorescens. The correlation between biogenic cyanide yield by P. putida and glycine concentration was proportional. In contrast, P. fluorescens showed a reduction in cyanide production at high glycine concentration, suggesting a potential feedback at glycine concentration above 8 g/L−1. Cyanide concentration was not detected in the control samples without glycine supplementation. The initial Au content in PCB sample measured by inductively coupled plasma mass spectrometry was 180 ppm in crushed PCB. In our two-step bioleaching approach, the remaining PCB from the Cu bioleaching reaction was added at the highest concentration of cyanide production in the P. putida and P. fluorescens cultures. The highest Au mobilization from discarded PCB reached by P. putida was 45%, whereas, P. fluorescens showed less Au mobilization ability i.e., nearer to 35% (Fig. 6). The addition of the AHL synthetic analogues C14H25NO3 and C16H27NO4 to the bioleaching reaction slightly increased the Au bioleaching capability of the two Pseudomonas strains. However, the effect was more evident in P. fluorescens (Fig. 6). Au was not found in the sterile control, indicating that biogenic cyanide was the only gold leaching agent present.

Figure 6 Gold bioleaching assays using P. putida. and P. fluorescens as bioleaching agents at glycine concentration of 12 g/L−1.

The effect of C14H25NO3 and C16H27NO4 over bioleaching ability was assessed. Each bar represents the average of three replicate samples. Comparable patterns of Au bioleaching were obtained in three independent experiments.

Discussion

The biohydrometallurgy method is a highly favorable approach (eco-friendly and cost effective) to solubilize several metals from e-waste. The microorganisms used in this process require metal elements for their structural and metabolic functions. A better understanding of the microbiological mechanisms involved in the bioleaching reaction could help overcome limitations at the industrial level. Quorum sensing is a cell to cell communication system that enables the bacterial community to respond in a concerted fashion to several environmental changes (Ng & Bassler, 2009). Because the QS system regulates approximately 25% of non-essential genes, its modulation could lead to the promotion and selection of particular phenotypical traits. In this study, a two-step bioleaching approach was performed using the bacteria A. ferrivorans and A. ferrooxidans in order to solubilize Cu, as well as the bacteria P. putida and P. fluorescens in order to solubilize Au from discarded PCB. Furthermore, the effect of the HLA Quorum Sensing system on the bioleaching reaction was assessed by adding long chain synthetic analogues. To the best of our knowledge, this is the first study using quorum sensing pathway modulation in order to enhance gold and copper bioleaching ability in urban mining.

In our study using 1% of ground PCB, 96% of the Cu was mobilized from the discarded PCBs with a mixed culture of A. ferrivorans and A. ferrooxidans, while the bioleaching reactions using pure cultures of the bacteria A. ferrivorans and A. ferrooxidans obtained a maximum mobilization of Cu of 95% and 91%, respectively. This suggests that there is no significant additive effect of the acidophilic bacteria used in this study, despite the fact that the time required to reach the maximum bioleaching capacity is reduced (Fig. 3). This could be explained by the conformation of a cooperative bioleaching behavior of the two strains in co-culture, in which they combine the attachment and non-attachment strategies, resulting in a consolidation of reaction space that will favor the Fe oxidation.

At crushed PCB loads greater than 2.5% the bioleaching efficiency dropped drastically. These findings were consistent with those reported in previous studies (Ilyas et al., 2007; Janyasuthiwong et al., 2016). These findings could be explained by the toxic nature of the organic fractions of PCB i.e., brominated flame retardants (BFRs) and polybrominated diphenyl ethers (PBDEs) on acidophile bacteria (Zhou et al., 2013). Another possible explanation of the reduction in bioleaching ability by A. ferrivorans and A. ferrooxidans at PCB loads greater than 2.5%, could be due to the alkaline nature of PCB, which causes the pH in the bioleaching reaction to rise to levels at which acidophile bacteria cannot thrive (Brandl, Bosshard & Wegmann, 2001). The pre-growth phase performed in this study (culture of A. ferrivorans and A. ferrooxidans in the absences of discarded PCB), helped to reduce the toxic effect of PCB on acidophile bacteria, as well as to reduce the lag growth stage favoring the bioleaching conditions.

Our findings that the Cu bioleaching efficiency of A. ferrooxidans and A. ferrivorans is higher when there is a biofilm formation on PCB particles (Fig. 5), could be explained by the fact that bioleaching by acidophile bacteria is the result of three bacterial mechanisms: (i) attachment, (ii) no attachment and (iii) cooperativity. These three mechanisms usually work synergistically (Rawlings, 2002). The attachment mechanism requires close contact between bacteria and PCB surfaces, thus bacteria could obtain electrons from it. This mechanism depends rigorously on bacterial production of extracellular polymeric substances (EPS), which are essential for the following biofilm formation. The no attachment mechanism is characterized by oxidation of Fe2+ to Fe3+, thus, Fe is the oxidizing agent to solubilize the Cu present in PCB. The cooperative mechanism is a combination of attachment and no attachment mechanisms. Here, the biofilm matrix, mainly composed of EPS, will form the reaction space that will favor the Fe oxidation (San Martin & Aguilar, 2019). Therefore, the Cu bioleaching from PCB will be eased by the biofilm formation. One explanation for how the addition of N-dodecanoyl-DL-homoserine lactone (C16H29NO3) to the bioleaching reaction increased the Cu solubilization from PCB (Figs. 4 and 5) could be that in the bacterium A. ferrooxidans, the AfeI/R quorum sensing system regulates important bacterial traits such as: EPS production, biofilm formation, ferrous iron oxidation, cell growth and sulfur oxidation. These bacterial traits are crucial for the bioleaching reaction. The encoding genes for AfeI/R proteins are over-expressed at the exponential growth stage (San Martin & Aguilar, 2019). Additionally, the biofilm provides a protective effect to the bacterial population that compose it, against harmful agents. Interestingly, the genome of A. ferrooxidans has ten clusters of genes predicted to belong to the resistance-nodulation-cell division (RND) family of transporters (Gao et al., 2021). Our results are consistent with those found in a previous study showing that the addition of synthetic homologues of AHL with long acyl chain improve the bioleaching activity in A. ferrooxidans (González et al., 2013).

In this study, the findings on Au bioleaching by Pseudomonas strains are consistent with those previously developed, in which a direct relationship between cyanide production and gold bioleaching capacity was reported (Pradhan & Kumar, 2012). The Au leaching is carried out by a process known as Cyanidation i.e., leaching metals with cyanide in alkaline conditions at pH >10.0 (Kunz et al., 1992). Biogenic cyanide leaching by bacteria shows similarities with chemical leaching, in terms of effectiveness. Nevertheless, the biogenic cyanide could be consumed by bacterial metabolism. In the bacterium P. fluorescens, cyanide is used as a nitrogen source and is subsequently converted to NH3 (Castric, 1977). This fact is consistent with our findings in which, P. fluorescens showed a reduction of nearly 15% in the Au bioleaching capacity compared to P. putida (Fig. 6). Finally, in the bacteria P. putida and P. fluorescens, HCN (hydrogen cyanide) biosynthesis is carried out by the enzyme HCN synthase. This enzyme is attached to membranes and catalyzes the formation of HCN and CO2 using glycine as a substrate (Castric, 1977). In Pseudomonas strains, a cluster of genes hcnABC encode HCN synthase. The hcnA promoter is positively regulated by the quorum sensing systems LasR and RhIR, which use the AHL molecule as an autoinducer (Villamizar et al., 2020). This fact explains the marked enhancement in Au bioleaching by P. fluorescens, after the addition of C14H25NO3 and C16H27NO4 to the bioleaching reaction (Fig. 6), as it was previously mentioned that P. fluorescens uses cyanide as a nitrogen source.

Conclusions

Bioleaching is a highly promising approach for urban biomining, especially in the field of metal recovery from e-waste. However, it is still not used sufficiently at an industrial level, due to the lack of knowledge of the microbiological mechanisms involved. In this study, the implementation of the two-step bioleaching approach was able to improve the efficiency of the bioleaching of Cu and Au from discarded PCBs. Furthermore, modulation of quorum sensing systems using synthetic analog auto-inducers achieved the promotion of bacterial phenotypic traits that enhance the bioleaching reaction. These could be potential tools to overcome the challenges on an industrial level.

Supplemental Information

Supplemental Information 1 Fig. 2 data

Click here for additional data file.

Supplemental Information 2 Fig. 3 Data

Click here for additional data file.

Supplemental Information 3 Fig. 4 Data

Click here for additional data file.

Supplemental Information 4 Fig. 5 Data

Click here for additional data file.

Supplemental Information 5 Fig. 6 Data

Click here for additional data file.

Supplemental Information 6 Raw data: measurements of the bioleaching assays and absorbance for biofilm determination assays

Click here for additional data file.

The authors thank to Emily McKinney for her critical reading and valuable suggestions.

Additional Information and Declarations

Competing Interests

Author Contributions

Data Availability

The authors declare there are no competing interests.

Juan Carlos Caicedo conceived and designed the experiments, performed the experiments, analyzed the data, prepared figures and/or tables, authored or reviewed drafts of the article, and approved the final draft.

Sonia Villamizar performed the experiments, analyzed the data, prepared figures and/or tables, authored or reviewed drafts of the article, and approved the final draft.

Giampaolo Orlandoni analyzed the data, authored or reviewed drafts of the article, and approved the final draft.

The following information was supplied regarding data availability:

The raw data are available in the Supplementary Files.

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
