# Peer review of "The use of synthetic agonists of quorum sensing N- acyl homoserine lactone pathway improves the bioleaching ability in Acidithiobacillus and Pseudomonas bacteria"

_PeerJ, doi:10.7717/peerj.13801_

## Round 0.1 · original submission · Major Revisions

Dear Dr. Caicedo and colleagues:

Thanks for submitting your manuscript to PeerJ. I have now received two independent reviews of your work, and as you will see, the reviewers raised some concerns about the research. Despite this, these reviewers are optimistic about your work and the potential impact it will have on research studying bacterial bioleaching approaches for e-waste remediation. Thus, I encourage you to revise your manuscript, accordingly, taking into account all of the concerns raised by both reviewers.

Importantly, please ensure that an English expert has edited your manuscript for grammar. Please also ensure that your figures and tables contain all of the information that is necessary to support your findings and observations. Statistical analyses are lacking and need to be provided. Please discuss your findings in light of relevant literature.

There are many minor suggestions to improve the manuscript.

Therefore, I am recommending that you revise your manuscript, accordingly, taking into account all of the issues raised by the reviewers.

Good luck with your revision,

-joe

Reviewer 1 ·

Basic reporting

Caicedo and Orlandoni describe the application of Acidithiobacillus and Pseudomonas bacterial species in leaching copper and gold, implementing N-AHLs to enhance metal leaching by quorum sensing. The bacterial species discussed in the paper have been recognized and characterized in copper/gold leaching previously. The addition AHL homologues facilitates metal bioleaching is also consistent with published paper. Authors discussed possible underlying mechanisms in the discussion session, but the paper doesn’t include mechanism study data to support the discussion, which weakens the novelty and profundity of this work.

Experimental design

no comment.

Validity of the findings

no comment.

Additional comments

Please provide high resolution images for figure 2 and 3.
In figure 5, A.ferrooxidans and A.ferrivorans are monitored under absorbance 562 nm. Please modified the “Material and Methods” session accordingly.

And the English language need to be improved to ensure international audience can understand the context clearly. Here are some examples where the language can be modified for easier understanding.
1. The second sentence in abstract. “However, the lack of complete understanding about microbiological processes involved in bioleaching reaction lead to drop in metal solubilization when using large amount of e-waste.
If “lead” is the verb of the sentence, then “leads” is the correct form.
If the original meaning is there is not enough understanding about how bioleaching reaction leads to drop in metal solubilization, then the sentence is not finished.
2. In the last sentence of abstract.
“Bio leaching reactions using bacterial quorum sensing modulation, it could represent a valuable tool… “ please remove “,” and it.
3. Similar issue in line 196-line198. Line 196-line 197 is not a sentence, please combine the line 198 with it.
4. There are many minor grammatical mistakes in the manuscript including:
Line 42: as well as it is also responsible for ….
“As well as” cannot be followed by a sentence.
Line 55-56: “… it leading to…”
“which leading to” is grammatically correct.
5. Many sentences start with ‘So,’ ‘Afterwards,’ ‘After,’ ‘Whoever,’. There is no need to add comma after these prepositions and adverbs. And sentences start with “it which” are grammatically incorrect.

Please have someone who is proficient in English and familiar with the subject of the manuscript to proofreading and editing the paper.

·

Basic reporting

The author reported the article according to the journal standard while there is some suggested improvement.

Experimental design

The experimental design is good however some of the improvement is needed in methodology section. I provided the comment in t

Validity of the findings

Novelty, rationale, conclusion are clearly defined

Additional comments

Following changes must be incorporated before publication in the journal
1. In Introduction section give more detail about bioleaching property of Acidithiobacillus and pseudomonas genus. Give detail of these bacteria on their source, living environment, genetic makeup, also state that these are pathogenic or non pathogenic (this is important when your research related to industry).
2. In methodology: Biofilm formation assays, How you observe the biofilm formation as the standard method for biofilm detection as scanning electron microscope. You can clear this point and include a picture of biofilm in the results section if applicable.
3. Include a statistical analysis at the end of method section under heading data analysis or statistical analysis
4. The author use Acidithiobacillus for copper bioleaching and pseudomonas for gold bioleaching. Explain why you not tested both the bacteria for copper and gold. If it is exclusion or inclusion criteria, mention it in methodology.
5. Previous studies are available on Acidithiobacillus and pseudomonas bioleaching. It should be better to explain the variation and your study strength compared to other studies in the Discussion section.

---

## Round 0.2 · accepted · Accept

Dear Dr. Caicedo and colleagues:

Thanks for resubmitting your manuscript to PeerJ. I now believe that your manuscript is suitable for publication. Congratulations! I look forward to seeing this work in print, and I anticipate it being an important resource for groups studying bacterial bioleaching approaches for e-waste remediation. Thanks again for choosing PeerJ to publish such important work.

Best,

-joe

Reviewer 1 ·

Basic reporting

The manuscript is well-written with professional English, and the experiment is clearly designed, and conclusion is solid.

Experimental design

N/A

Validity of the findings

N/A

Additional comments

N/A

·

Basic reporting

no comment

Experimental design

no comment

Validity of the findings

no comment

Additional comments

no comment